# 18Beta-Glycyrrhetinic Acid Attenuates H_2_O_2_-Induced Oxidative Damage and Apoptosis in Intestinal Epithelial Cells via Activating the PI3K/Akt Signaling Pathway

**DOI:** 10.3390/antiox13040468

**Published:** 2024-04-16

**Authors:** Cui Ma, Fuxi Wang, Jiawei Zhu, Shiyi Wang, Yaqing Liu, Jianfang Xu, Qingyu Zhao, Yuchang Qin, Wei Si, Junmin Zhang

**Affiliations:** 1State Key Laboratory of Animal Nutrition and Feeding, Institute of Animal Sciences of Chinese Academy of Agricultural Sciences, Beijing 100193, China; 82101211103@caas.cn (C.M.); zhaoqingyu@caas.cn (Q.Z.);; 2College of Animal Science and Technology, Shanxi Agricultural University, Jinzhong 030801, China; 3College of Food Science and Engineering, Qingdao Agricultural University, Qingdao 266109, China; 4College of Animal Science and Technology, Qingdao Agricultural University, Qingdao 266109, China

**Keywords:** 18beta-glycyrrhetinic acid, oxidative stress, PI3K/Akt pathway, molecular docking

## Abstract

Oxidative stress causes gut dysfunction and is a contributing factor in several intestinal disorders. Intestinal epithelial cell survival is essential for maintaining human and animal health under oxidative stress. 18beta-Glycyrrhetinic acid (GA) is known to have multiple beneficial effects, including antioxidant activity; however, the underlying molecular mechanisms have not been well established. Thus, the present study evaluated the therapeutic effects of GA on H_2_O_2_-induced oxidative stress in intestinal porcine epithelial cells. The results showed that pretreatment with GA (100 nM for 16 h) significantly increased the levels of several antioxidant enzymes and reduced corresponding intracellular levels of reactive oxidative species and malondialdehyde. GA inhibited cell apoptosis via activating the phosphoinositide 3-kinase/protein kinase B (PI3K/Akt) signaling pathway, as confirmed by RNA sequencing. Further analyses demonstrated that GA upregulated the phosphorylation levels of PI3K and Akt and the protein level of B cell lymphoma 2, whereas it downregulated Cytochrome c and tumor suppressor protein p53 levels. Moreover, molecular docking analysis predicted the binding of GA to Vasoactive intestinal peptide receptor 1, a primary membrane receptor, to activate the PI3K/Akt signaling pathway. Collectively, these results revealed that GA protected against H_2_O_2_-induced oxidative damage and cell apoptosis via activating the PI3K/Akt signaling pathway, suggesting the potential therapeutic use of GA to alleviate oxidative stress in humans/animals.

## 1. Introduction

The mammalian intestine, a complex and highly regulated organ, is responsible for several key functions, including nutrient digestion and absorption, serving as a protective barrier against invasion by bacteria, and endocrine regulation through blood circulation [1]. Owing to its complex physiological and/or chemical environment, the intestine is susceptible to diverse stress injuries. Intestinal oxidative stress leads to various inflammatory bowel diseases such as ulcerative colitis, Crohn’s disease, and indeterminate colitis [2]. The impaired epithelial integrity of the intestine can even cause damage to external organs [3].

Oxidative stress is an imbalance between oxidation and antioxidation, which results in the excessive production of reactive oxygen species (ROS) [4]. High exposure to ROS causes cell membrane damage and apoptosis and is one of the major factors leading to cell apoptosis. For instance, increased ROS levels in alcohol-exposed rats or humans led to decreased levels of tight junction (TJ) and adhesion (AJ) proteins in enterocytes, thereby accelerating apoptosis [5]. It has been reported that oxidative damage could upregulate pro-apoptotic proteins such as Cytochrome c (Cyt c) and BCL2-associated X (BAX) to promote cell apoptosis [6]. Moreover, oxidative stress causes intestinal stem cell and intestinal barrier damage in pigs by decreasing the expression of tight junction proteins [7]. Increased intestinal permeability can lead to piglet diarrhea, impair immune function, as well as reduce the growth performance of piglets [8]. Therefore, effective and safe nutritional approaches to prevent oxidative stress-induced cell apoptosis are of urgent need.

Licorice extract exhibits a range of potential biological activities, including antibacterial, anti-inflammatory, antiviral, antioxidant, and antidiabetic effects [9]. The main active component with antioxidant properties in licorice is 18beta-glycyrrhetinic acid (GA), which is a potential candidate for the prevention and/or treatment of oxidative stress. A previous study reported that GA improved kidney oxidative stress injury through upregulating endogenous antioxidants [10]. Additionally, it can protect the liver from oxidative injury by inhibiting free-radical production and lipid peroxidation [11]. Moreover, GA can also increase antioxidant enzyme levels and Bcl-2 expression to alleviate oxidative stress in rat liver [12]. However, the underlying mechanisms by which GA protects against damage caused by oxidative stress have not been explored in depth.

Therefore, we hypothesized that GA could be a potential candidate for the prevention of oxidative stress. Thus, the present study evaluated the protective effects of GA in oxidative stress-induced apoptosis and identified its molecular mechanisms. We established an in vitro oxidative stress model induced by hydrogen peroxide (H_2_O_2_) using a porcine intestinal epithelial cell line (IPEC-J2). We demonstrated that GA protected IPEC-J2 cells from H_2_O_2_-induced oxidative stress by increasing cell viability and decreasing ROS production and cell apoptosis. GA directly bound to a master membrane receptor Vasoactive intestinal peptide receptor 1 (VIPR1) as predicted by a molecular docking assay, and thereby activated phosphoinositide 3-kinase/protein kinase B (PI3K/Akt) signaling. We also confirmed that GA pretreatment triggered the increase in B cell lymphoma 2 (BCL2) and decreased Cyt c and tumor suppressor protein p53 (p53), the key proteins involved in cell apoptosis. Our findings indicated that GA could be used as a potential agent for the prevention of diseases related to oxidative stress, such as acute/chronic intestine and/or liver disease.

## 2. Materials and Methods

### 2.1. Cell Culture

Porcine IPEC-J2 cells were maintained in DMEM/F12 (GIBCO, Grand Island, NY, USA) medium containing 10% (*v*/*v*) fetal bovine serum and 1% penicillin (100 IU/mL)/streptomycin (100 µg/mL). IPEC-J2 cells were cultured in a controlled atmosphere of 95% humidity with 5% CO_2_ at 37 °C.

### 2.2. Detection of IPEC-J2 Cell Viability

IPEC-J2 cells were seeded in a 96-well plate at a density of 2 × 10^4^ cells per well and cultured for 24 h (reaching 70–80% confluence). GA (Yuanye, Shanghai, China) was dissolved in dimethyl sulphoxide (DMSO, purity ≥ 99%) and incubated with IPEC-J2 cells at various concentrations (50 nM, 100 nM, 500 nM, and 2 μM) and times (12 h, 24 h, and 48 h) according to previous studies [13,14,15]. Then, 200 μM of H_2_O_2_ was added to cells for 8 h. Subsequent procedures were carried out according to the instructions of a Cell Counting Kit-8 (CCK8) kit, and 10 μL of CCK-8 reagent was added to each well for 4 h at 37 °C. Finally, the absorbance value was measured at 450 nm using an Infinite F50 plate reader (Tecan, Männedorf, Switzerland).

### 2.3. Assessment of Cellular Barrier

IPEC-J2 cells were seeded in six-well plates (5 × 10^5^ cells per well) for 24 h, then treated with 100 nM of GA for another 24 h. Then, 200 μM of H_2_O_2_ was added for the last 8 h. The experimental treatment included the control (CK), H_2_O_2_-exposed (H_2_O_2_), pretreatment with GA for 24 h and exposure to 200 μM of H_2_O_2_ for 8 h (GA + H_2_O_2_), and pretreatment with GA for 24 h (GA) groups. The activities of lactate dehydrogenase (LDH, Jiancheng, Nanjing, China), diamine oxidase (DAO, Jiangsu Meimian Industrial Co., Ltd., Yancheng, China), and the level of D-lactic acid (D-LA, Jiancheng, Nanjing, China) in the cell culture supernatant were determined using corresponding assay kits.

### 2.4. Apoptosis Analysis

IPEC-J2 cells were cultured in six-well plates at a concentration of 5 × 10^5^ cells per well to attach for 24 h and then treated with 100 nM of GA for another 16 h. Following the GA treatment, IPEC-J2 cells were incubated with H_2_O_2_ for 8 h. The cells were digested with trypsin without ethylenediamine tetraacetic acid (EDTA), and the apoptosis was performed with an Annexin V-Fluorescein isothiocyanate/propidium iodide (FITC/PI) apoptosis detection kit (Invitrogen, Carlsbad, CA, USA) following the manufacturer’s protocols. The cells obtained were washed three times with PBS and centrifuged at 3000× *g* for 5 min at room temperature. Then, 200 μL of 1 × binding buffer and 5 μL of Annexin V-FITC were added to 100 μL of IPEC-J2 cell suspension at room temperature in a dark place for 15 min. Finally, 200 μL of 1 × binding buffer and 5 μL of PI staining solution were added before analysis (within 1 h) using a flow cytometer.

### 2.5. Intracellular ROS Levels

Intracellular ROS accumulation was measured with the fluorescent probe 2′,7′-dichlorofluorescein diacetate (DCFH-DA; MedChemExpress, Monmouth Junction, NJ, USA) as described previously [16]. IPEC-J2 cells were seeded in six-well cell culture plates, treated with 100 nM of GA for 24 h, and exposed to 200 μM of H_2_O_2_ after 8 h incubation. The cells were treated with 10 μM of DCFH-DA for 30 min at 37 °C. After washing with PBS, cells were visualized and photographed under a fluorescence microscope (Leica TCS SP8, Leica Microsystem, Hessen, Wetzlar, Germany). To evaluate ROS production, the fluorescence intensity was measured using a fluorescence microplate reader (Infinite 200 Pro, Tecan, Männedorf, Switzerland) at excitation/emission wavelengths of 485/530 nm. Finally, fluorescence intensities were normalized with protein concentrations.

### 2.6. Redox Status Analysis

The cell and supernatant were collected to detect the malondialdehyde (MDA), superoxide dismutase (SOD), and glutathione peroxidase/oxidized glutathione disulfide (GSH/GSSG) rate, as well as the catalase (CAT) and superoxide anion radical scavenging activities, with commercial assay kits (Appendix A) purchased from Nanjing Jiancheng Bioengineering Institute (Nanjing, China), according to the manufacturer’s instructions. Briefly, the MDA level was measured based on the chemical reactivity of MDA with thiobarbituric acid (TBA), resulting in the formation of a distinctive red product, which was quantitatively assessed at 532 nm. The SOD activity was measured based on the principle that the byproducts (superoxide anions) during xanthine oxidase-catalyzed xanthine oxidation oxidize hydroxylamine to nitrite, resulting in an amaranth purple color. The developed color was subsequently measured at 550 nm. The total GSH is measured by reacting GSH and GSSG with 5,5′-dithiobis (2-nitrobenzoic acid) (DTNB) to yield 5-thio-2-nitrobenzoic acid (λ = 412 nm). Derivatized GSH with 2-vinylpyridine and only GSSG undergoes the DTNB reaction, and the GSSG level was measured at 412 nm. The GSH level was determined using the formula GSH = Total GSH − 2GSSG, and the ratio of GSH/GSSG was calculated. CAT reacts with H_2_O_2_, and this reaction can be halted by molybdenum to yield a yellow product. The activity of CAT was assessed by monitoring the reduction in absorbance at 405 nm, attributed to the degradation of H_2_O_2_.

### 2.7. RNA-Seq Library Construction and Analysis

The total RNA was assessed using the RNA Nano 6000 Assay Kit of the Bioanalyzer 2100 system (Agilent Technologies, Santa Clara, CA, USA). RNA-seq was conducted by the Novogene Company (Tianjin, China). The library fragments were purified with an AMPure XP system (Beckman Coulter, Beverly, MA, USA). The PCR product was purified by using AMPure XP beads quantified by a Qubit2.0 Fluorometer and detected by an Agilent 2100 bioanalyzer to obtain the final library. Subsequently, the relevant data were sequenced on an Illumina NovaSeq 6000. High-quality RNA samples were further purified using the rRNA-depletion method and used to construct strand-specific RNA-seq libraries. The reads that matched perfectly or only had one single mispairing were analyzed further. Fragments per kilobase of transcript sequence per million mapped reads (FPKM) were used to determine differentially expressed genes (DEGs), and the DEGs were identified with the DESeq2 analysis procedure. Volcano plots were established by comparing log_10_ (statistical relevance) to log_2_ (fold change). Pathway enrichment analyses were used to identify functional classification or metabolic pathways enriched by the DEGs. Raw sequence data are available in the NCBI Sequence Read Archive with accession number PRJNA1005228.

### 2.8. Quantitative Real-Time PCR Assay

The quantitative real-time PCR was performed as described previously [17]. Briefly, the total RNA was extracted, and cDNA was reverse transcribed from 1 μg of RNA using a reagent kit following the manufacturer’s instructions. An RT-qPCR analysis was performed on an ABI 7500 Real-Time PCR system (Applied Biosystems, Foster City, CA, USA) with TB Green Premix Ex Taq II (Tli RNaseH Plus). The primer sequences used in this study are listed in Appendix A. The 2^−ΔΔCT^ method was used to calculate the gene expression levels of the target genes [18]. The relative mRNA expression levels were normalized with the reference gene *GAPDH*. Each gene was analyzed in triplicate.

### 2.9. Western Blot Analysis

The cultured IPEC-J2 cells were harvested to analyze protein abundance as described previously [19]. Briefly, the protein was extracted from cells by RIPA protein lysate containing protease inhibitor cocktail and phosphatase inhibitor cocktail. The concentrations of protein samples were detected with a bicinchoninic acid (BCA) protein assay kit following the manufacturer’s protocols. Protein samples (30 μg) were separated on SDS-PAGE gels and transferred to polyvinylidene difluoride membranes. After being blocked for 1 h, cell membranes were incubated with primary and secondary antibodies. The detailed information on all the antibodies used is listed in Appendix A. Finally, protein abundance was detected and quantified with an automatic chemiluminescence image analysis system (Tanon 5200, Hangzhou, China) and ImageJ 5.1 software. The expression levels of proteins were normalized with β-Actin.

### 2.10. Immunofluorescence and Confocal Microscopy

For the immunofluorescence study of p-Akt, IPEC-J2 cells were plated on glass slides in 24-well plates at a density of 1 × 10^5^ cells/well for 24 h and then treated with 100 nM of GA for 16 h followed by 200 μM of H_2_O_2_ for 8 h. Subsequently, the plates with slides were washed in PBS for 5 min, followed by three PBS washes. The slides were fixed with 4% paraformaldehyde at 37 °C for 1 h and then washed with PBS for 5 min, followed by three PBS washes. Cells were permeabilized with 0.5% Triton X-100 (prepared with PBS) for 30 min at room temperature and washed in PBS for 5 min, followed by three PBS washes, and then blocked with 10% goat serum prepared with PBS (Sigma Aldrich, St. Louis, MO, USA) at 37 °C for 1 h. Then, the p-Akt polyclonal rabbit antibody (9271, Cell Signaling Technology, Danvers, MA, USA) and Alexa Fluor 647 goat anti-rabbit (4414S, Cell Signaling) were used at 1:100 in PBS with 5% goat serum. Cells were incubated with the primary antibody p-Akt overnight at 4 °C and washed again, followed by incubation with fluorescent-conjugated secondary antibodies at 37 °C for 1 h. Then, cells were washed again and incubated with 0.5 μg/mL of 2-(4-amidinophenyl)-6-indolecarbamidine dihydrochloride (DAPI) (1:2000 in PBS) for 10 min at room temperature, washed, and sealed with sealing liquid containing an anti-fluorescence quencher. Then, the observed images were collected under a fluorescence microscope using LAS-X software 1.4.6 (Leica TCS SP8, Leica Microsystem, Hessen, Wetzlar, Germany). The mean of the fluorescent intensity of the channels in each cell was measured using LAS-X for quantification. Images were processed using ImageJ, and graphs were generated by GraphPad Prism 8.0.

### 2.11. Molecular Docking

The binding mechanisms and activity between active ingredients and abundant target proteins were predicted by the molecular docking between small molecules and relevant targets [20]. To predict the interaction sites of GA with VIPR1 and PI3K, the 3D structure of GA (PDB ID: RES_d_1) from the PubChem database (https://pubchem.ncbi.nlm.nih.gov/ (accessed on 12 June 2023)) was converted from the SDF to the mol2 format using Chem 3D software 4.0. The PDB format structures of the VIPR1 (UniProt: Q28992) and PI3K (PDB ID: 1E7U) were downloaded from UniProt and the Protein Data Bank and then the interaction of GA with VIPR1 and PI3K was analyzed using Autodock 1.5.6 and PyMOL 3.0 as described previously [21].

### 2.12. Statistical Analysis

All the data obtained from cell experiments were analyzed by one-way ANOVA and multiple comparisons were conducted with Duncan’s multiple range test using the SPSS 22.0 statistical package (SPSS Inc., Chicago, IL, USA). Images were generated with GraphPad Prism 7. All analyses were performed with three independent experiments. All data are expressed as the means ± standard error of the mean (SEM). The values were considered statistically significant when *p* < 0.05.

## 3. Results

### 3.1. GA Restores H_2_O_2_-Induced Cell Viability in IPEC-J2 Cells

Porcine IPEC-J2 cells were incubated with 200 μM of H_2_O_2_ for 8 h of induced intracellular oxidative stress [22,23,24]. We investigated the effect of exposure to 200 μM of H_2_O_2_ on the viability of IPEC-J2 cells using CCK-8 assays. We found that cell viability was significantly reduced to 70% after 8 h of incubation with 200 μM of H_2_O_2_ (*p* < 0.05, Appendix A). Notably, GA did not affect the cell viability of IPEC-J2 cells when the concentration ranged from 50 nM to 2 μM with different incubation times (12 h, 24 h, and 48 h, respectively) (*p* > 0.05, Figure 1A). Next, we pretreated the IPEC-J2 cells with different doses of GA for 24 h to test the protective effect. As shown in Figure 1B, GA pretreatments restored the viability of oxidative stress-induced cells to the control levels (*p* > 0.05, Figure 1B). Notably, 24 h of incubation with 100 nM of GA significantly increased the cell viability both in the control and oxidative stress-induced IPEC-J2 cells (*p* < 0.05, Figure 1A,B). Thus, we treated IPEC-J2 cells with 100 nM of GA for 24 h for the remaining analyses in this study.

### 3.2. GA Protects IPEC-J2 Cells from Cell Injury and Apoptosis under Oxidative Stress

To determine whether GA benefits plasma membrane integrity, we measured the release of LDH in the culture supernatant of IPEC-J2 cells. The LDH release in H_2_O_2_-treated cells was increased by 65.86% compared with the control cells (*p* < 0.05, Figure 2A). Pretreatment with GA (100 nM) significantly attenuated the H_2_O_2_-induced increase in LDH release (*p* < 0.05, Figure 2A). However, no significant difference was observed in LDH release between the control and GA-treated group, signifying its safety (*p* > 0.05, Figure 2A). In addition, we detected the levels of d-lactic acid (D-LA) and DAO, two critical parameters reflecting the intestinal barrier function and degree of damage. Both D-LA and DAO levels/concentrations were significantly increased upon H_2_O_2_ treatment, while they were markedly decreased by pretreatment with GA (*p* < 0.05, Figure 2B,C). These findings indicated that GA improved the protective capacity of the cell plasma membrane.

To further explore the protective effect of GA on apoptosis, we conducted an Annexin V-FITC/PI assay by flow cytometry. A higher percentage of apoptotic cells (7.15%) was observed with H_2_O_2_ treatment compared with the control group (4.60%) (Figure 2D,E). GA pretreatment significantly inhibited H_2_O_2_-induced cell apoptosis (3.32%) (Figure 2D,E). These findings provided further evidence that GA prevented IPEC-J2 apoptosis and improved cell viability.

### 3.3. GA Reduces Cellular ROS Production and Increases Antioxidant Enzyme Secretion in IPEC-J2 Cells

To investigate whether the observed attenuation in apoptosis resulted from the antioxidative effect of GA, we conducted a DCFH-DA assay to measure the intracellular ROS production in IPEC-J2 cells. Exposure to H_2_O_2_ (200 μM) alone triggered a substantial ROS production in IPEC-J2 cells, as observed by an increase in green dots under the microscope and higher quantified green fluorescence intensity (Figure 3A,B), whereas pretreatment with GA (100 nM) reduced the level of H_2_O_2_-induced ROS production compared to the control (*p* < 0.01, Figure 3B). However, GA pretreatment did not affect ROS production in IPEC-J2 cells (Figure 3A,B). These data suggested that GA protected IPEC-J2 cells against H_2_O_2_-induced oxidative stress, partially at least, due to its ability to scavenge or inhibit ROS.

High ROS levels induce iNOS expression [25]. Inspired by this, we detected the mRNA expression of *iNOS* in IPEC-J2 cells and found that H_2_O_2_ significantly increased the mRNA level of *iNOS*, whereas GA pretreatment inhibited the H_2_O_2_-induced *iNOS* level (*p* < 0.05, Figure 3C). Exposure to 200 μM of H_2_O_2_ elevated MDA levels and reduced the GSH/GSSG ratio, whereas pretreatment with GA significantly reversed the changing trend of both the MDA level and the GSH/GSSG ratio induced by H_2_O_2_ (*p* < 0.05, Figure 3D,E). These results suggested that GA alleviated the oxidative stress induced by H_2_O_2_ in IPEC-J2 cells. Next, we investigated the effect of GA on H_2_O_2_-induced antioxidant enzyme activity. As shown in Figure 3F,G, SOD and CAT activities were increased with GA pretreatment compared with the group exposed to H_2_O_2_ alone (*p* < 0.05). Hydrogen peroxide, a byproduct of cell metabolism, can be degraded by CAT, which prevents it from producing damaging hydroxyl radicals. Hydroxyl radical scavenging activity was decreased in the IPEC-J2 cells exposed to 200 μM of H_2_O_2_ (*p* < 0.05), whereas GA pretreatment restored the hydroxyl radical scavenging activity in IPEC-J2 cells (*p* < 0.05, Figure 3H). Taken together, these results indicated that GA promoted the secretion of antioxidant enzymes in response to oxidative stress-induced damage.

### 3.4. GA Alleviates H_2_O_2_-Induced Oxidative Stress by Activating the PI3K/Akt Pathway

RNA-seq was performed to investigate the molecular mechanisms by which GA alleviates H_2_O_2_-induced apoptosis and oxidative stress in IPEC-J2 cells (Appendix A). The activation of the PI3K-Akt pathway, one of the most well-known ROS-regulated pathways, is known to inhibit cell apoptosis and promote cell survival [26]. Thus, we focused on the effects of GA on the PI3K-Akt pathway in IPEC-J2 cells. Ceruloplasmin (CP) inhibits lipid oxidation. Additionally, the mRNA expression of human proline-rich 5 like (*PRR15L*) and zinc finger protein 554 (*ZNF554*) was negatively correlated with apoptosis [27,28]. VIPR1 (VPAC1) can exert anti-inflammation or antioxidant activity. The MDA level induced by H_2_O_2_ was decreased through activating VIPR1 [29]. The PI3K/Akt pathway is activated to maintain physiological homeostasis by the Gβ subunit when VIPR1 (VPAC1 or PACAP) of the class B GPCRs is activated [30,31]. Pretreatment with GA and exposure to H_2_O_2_ significantly increased (*p* < 0.05) the mRNA expression of *CP*, *ZNF554*, and *PRR15L* compared with the group exposed to H_2_O_2_ alone (Figure 4A). H_2_O_2_-induced oxidative stress significantly decreased (*p* < 0.05) the mRNA expression of *CP* and *VIPR1* compared with the control group (Figure 4B). These results indicated that GA probably activated the PI3K/Akt pathway in IPEC-J2 cells.

Western blot analysis was conducted to detect PI3K (p-PI3K) and Akt (p-Akt) phosphorylation levels in IPEC-J2 cells treated with GA without H_2_O_2_. However, GA alone did not affect the levels of either p-PI3K or p-Akt during steady-state conditions (without H_2_O_2_ exposure) (Figure 5A–C). Notably, under oxidative stress (with H_2_O_2_ exposure), GA pretreatment significantly increased (*p* < 0.05) the abundance of p-PI3K/PI3K and p-Akt/Akt, suggesting the activation of the PI3K/Akt signaling pathway (GA + H_2_O_2_ versus H_2_O_2_, Figure 5B,C). Furthermore, we characterized the subcellular localization of p-AKT by IF staining. GA pretreatment increased the abundance of p-Akt, mostly in the nucleus of H_2_O_2_-exposed IPEC-J2 cells (Figure 5D). Thus, it was conceivable that the protective role of GA in oxidative-stressed IPEC-J2 cells required the PI3K/Akt pathway.

To test our concept further, we assessed the effect of GA on the expression of several pro- and anti-apoptotic proteins in H_2_O_2_-exposed IPEC-J2 cells. The Western blotting results showed that H_2_O_2_ caused a reduction in the expression of the anti-apoptotic protein BCL-2, whereas pretreatment with GA significantly attenuated BCL-2 expression (Figure 6A,B). The expression of BAX was not affected by either H_2_O_2_ or GA; the BCL2/BAX ratio remarkably decreased in the H_2_O_2_-treated cells and considerable elevation was observed when pretreated with GA (Figure 6A,D). In addition, the levels of p53 and Cyt c were increased by H_2_O_2_, whereas GA pretreatment reduced the p53 and Cyt c levels in IPEC-J2 cells (Figure 6A,E,F). Furthermore, we observed a consistent change in p53 and Cyt c mRNA levels by qPCR analysis (Figure 6G,H), suggesting that GA has an anti-apoptotic role in H_2_O_2_-exposed IPEC-J2 cells. Collectively, our findings indicated that GA alleviated IPEC-J2 cell apoptosis by activating the PI3K/AKT pathway under oxidative stress.

### 3.5. Molecular Binding between GA and Potential Target Proteins

To investigate the binding modes between GA and the potential targets of the PI3K/Akt pathway, a molecular docking simulation was performed. This is a method for studying the interaction and recognition between proteins and ligands. It is well known that the binding of a ligand to PACAP receptors (VIPR1 and VIPR2), master membrane receptors, activates the main second messengers of the PI3K/Akt pathway [31]. Considering that GA increased the mRNA expression of VIPR1 in IPEC-J2 cells, as indicated by both RNA-seq and RT-qPCR analysis, we reasoned that GA activates intracellular PI3K through VIPR1. The molecular docking result showed that GA binds to VIPR1, forming one or two hydrogen bonds with the protein residues (Figure 7A). Hydroxyl groups at C-3 and/or C-30 formed hydrogen bonds with VIPR1 (Arg 189 and Lys 196) (Table 1). Next, we explored whether GA directly binds to PI3K. We found that hydrogen bonds were formed between GA and PI3K at C-3 and/or C-30 sites (Figure 7B, Table 1). It is generally accepted that strong stable ligand–receptor binding correlates with a greater binding energy value and better docking [32]. The docking results showed that the binding energy between GA and VIPR1 was greater than that between GA with PI3K, suggesting that GA bound to the VIPR1 receptor, and thereby activated intracellular PI3K/Akt signaling to alleviate oxidative stress in IPEC-J2 cells.

## 4. Discussion

In this study, pretreatment with 100 nM of GA decreased the production of ROS in IPEC-J2 cells and increased the activity of antioxidant enzymes. GA was observed to have similar protective effects against ROS-induced oxidative stress in previous studies [33,34]. Other studies have reported that ROS-induced oxidative cellular damage is usually accompanied by increased LDH activity, thereby reducing membrane permeability and antioxidant activity [35,36]. We found that pretreatment with GA remarkably increased LDH activity to improve cell barrier integrity. We also found that GA decreased H_2_O_2_-induced cell apoptosis, and pretreatment with 100 nM of GA could reverse the reduction of cell viability in IPEC-J2 cells. Therefore, GA has the potential to protect intestinal barrier function and restore cell viability under oxidative stress.

An enhanced cell antioxidant defense system leads to ROS scavenging, thus alleviating oxidative stress. As one of the most important products of membrane lipid peroxide, MDA aggravates cell damage [37,38]. Superoxide anions, a relatively weak oxidant, trigger lipid peroxidation and lead to cell DNA damage and death [39]. CAT and glutathione peroxidase are known as the main antioxidant enzymes in cells [40,41]. A previous study reported that GA might increase the CAT activity to scavenge oxygen free radicals and verified this experimentally by the use of molecular docking [42]. To explore the effects of GA on antioxidant activity in vitro, we established an oxidative stress model using IPEC-J2 cells induced by H_2_O_2_ and found that GA exhibited promising in vitro O_2_^•−^ scavenging activity and reduced the levels of ROS-induced cell lipid peroxide. However, GA supplementation did not increase the antioxidant-related gene expressions by RNA-seq. The discrepancies between mRNA expression and protein levels have also been observed by others. Röhrdanz et al. showed that the activity of antioxidant enzymes (such as CAT and SOD) was inconsistent with mRNA expression [43]. Xu et al. reported that an increase in MnSOD mRNA expression was not consistent with the increase in MnSOD activity [44]. These suggested that a translational block might exist for the antioxidant enzyme protein. It had been proved that the difference in protein level versus transcript level changes was controlled by a post-transcriptional modification mechanism [45].

The PI3K/Akt pathway plays an important role in oxidative stress [46]. In the present study, the p-Akt protein level was increased in the GA + H_2_O_2_ group, and pretreatment with GA significantly increased the p-Akt protein level compared with oxidative stress alone. We also found that pretreatment with GA could promote the expression level of Akt in the nucleus. Pretreatment with GA significantly decreased the protein levels and gene expression of Cyt c and p53, increasing the ratio of BCL2/BAX. These indexes are known to be related to the PI3K/Akt pathway. As a downstream effector of PI3K, Akt can inhibit oxidative stress by regulating downstream target proteins of the Bcl-2 family [47]. In addition, the Bcl-2 family of proteins controls mitochondrial permeability. Moreover, the Bcl-2 protein resides in the outer membrane of mitochondria to inhibit cytochrome release [48,49]. Oxidative stress may promote the leakage of Cyt c from mitochondria to cytoplasm by changing mitochondrial membrane permeability. In addition, Ghatei et al. reported that p53 promotes oxidative stress injury by downregulating the expression of BCL2 and decreasing the BCL2/BAX ratio [50]. These findings suggest that GA alleviates oxidative stress by activating the PI3K/Akt pathway. Oxidative stress might also interfere with the activity of Akt in the PI3K/Akt pathway by disrupting the oxidative redox homeostasis of cells [51]. Thus, we postulated that oxidative stress promoted p-Akt phosphorylation but affected its activity. However, it was unclear how GA may promote Akt activity. Thus, further in-depth studies are necessary to reveal the exact relationship between Akt activity and oxidative stress.

The small drug molecule binds to an amino acid residue on the receptor to form a drug–receptor complex. Small drug molecules can activate or inhibit the bioactivities of the receptors mainly by the use of intermolecular electrostatic interactions and ionic bonds, hydrogen bonds, and Van der Waals forces [52,53]. VIPR1 is a member of class B receptors belonging to the super-family of GPCR. Previous studies showed that the N-terminal ectodomain (N-ted) of the VPAC1 receptor plays a crucial role in small molecule agonist binding [54]. VIPR1 plays a crucial role in cell membrane functions. VIPR1 was activated by a ligand and further to activate the cAMP/PKA pathway to inhibit inflammation and oxidative stress, suggesting that VIPR1 was mainly used as a membrane receptor for signal transduction [55,56]. IPR1 could initiate a series of cellular regulatory processes to influence the physiological state of the cell, such as cell proliferation, differentiation, and metabolism [57]. However, further in-depth studies are necessary to explore the cellular role of VIPR1. We found that GA could bind to the VIPR1 membrane receptor by molecular docking and improve the viability of IPEC-J2 cells. However, the binding of GA and VIPR1 would be validated for further investigations, such as the Western blot experiment as well as CETSA and radioligand binding experiments at the protein level, and the siRNA experiment would be performed at the cellular level. VIPR1 has tyrosine kinase domains that contribute to the development of cell growth [58]. Previous studies have reported that PI3K/Akt was activated downstream to improve cell survival when the VIPR1 receptor of the cell membrane was activated [59,60]. This implied that pretreatment by GA activated VIPR1 in the membrane to upregulate PI3K/Akt and inhibit cell apoptosis in IPEC-J2 cells. Previous studies have found that GA reduced oxidative stress injury by activating the PI3K/Akt signaling pathway [61]. It is speculated that GA may directly promote the translocation of Akt to the membrane by binding to PI3K, thereby activating Akt and promoting its expression. These findings indicate that GA acts as a ligand and binds to VIPR1 in the cell membrane, thereby activating the intracellular PI3K/Akt signaling pathway to attenuate cell viability under oxidative stress. However, further studies are needed to explore the relationship between VIPR1 and PI3K in vitro and in vivo.

## 5. Conclusions

In summary, pretreatment with GA reduced oxidative stress-induced cell apoptosis by inhibiting the production of intracellular ROS, increasing antioxidant enzyme levels, and decreasing apoptotic protein/gene expression. Furthermore, GA pretreatment activated the PI3K/Akt signaling pathway by binding to the VIPR1 of the outer cell membrane and then activating Akt in the inner cell membrane, thereby improving the viability of IPEC-J2 cells impaired by oxidative stress. The findings in the present study provided evidence for the development of GA as a prophylactic and/or therapeutic agent for diseases related to oxidative stress, such as aging and liver disease. Notably, our findings also provided data for the future application of GA as an antibiotic alternative to improve the intestinal health of animals.

## Figures and Tables

**Figure 1 antioxidants-13-00468-f001:**
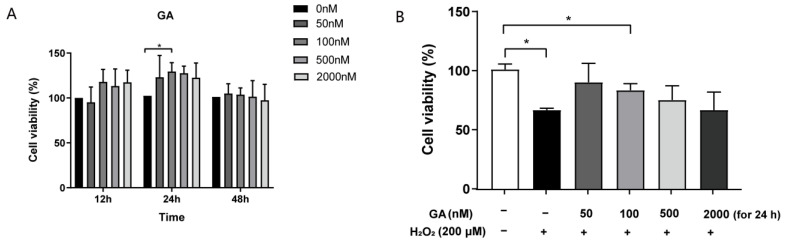
Effect of glycyrrhetinic acid (GA) on the cell viability of IPEC-J2 cells exposed to oxidative stress. (**A**) Effect of GA on cell viability of IPEC-J2 cells. Cells were treated with GA 50 nM, 100 nM, 500 nM, and 2 μM GA for 12, 24, and 48 h, respectively. (**B**) Cells were treated with GA 50 nM, 100 nM, 500 nM, and 2 μM GA for 24 h, and exposed to 200 μM H_2_O_2_ for 8 h. (**A**,**B**) Data were analyzed by one-way ANOVA, and multiple comparisons were conducted with Duncan’s multiple range test. Data are presented as mean ± SEM with three independent experiments. n = 3 means three independent experiments. The differences between two groups were considered significant at * *p* < 0.05.

**Figure 2 antioxidants-13-00468-f002:**
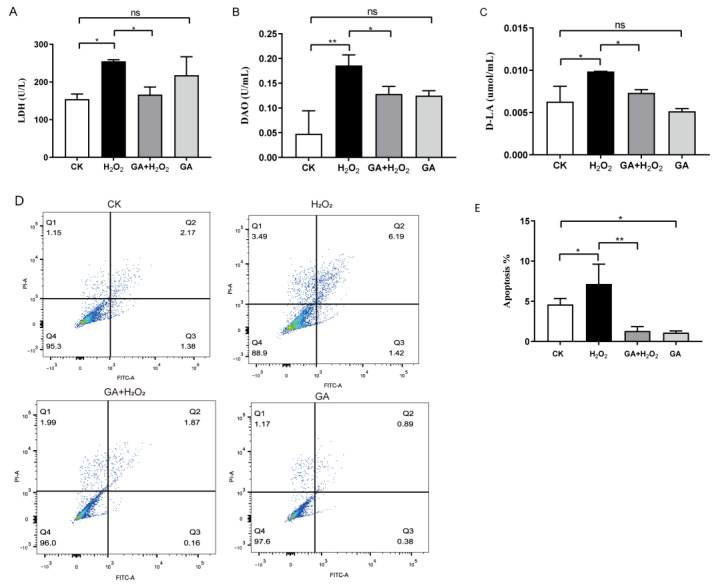
Glycyrrhetinic acid (GA) prevents oxidative stress-induced IPEC-J2 cell barrier damage and apoptosis. Cells were seeded for 24 h, pretreated with 100 nM GA for 16 h, then exposed to 200 μM H_2_O_2_ for 8 h. (**A**–**C**) Effects of GA on lactate dehydrogenase (LDH) and diamine oxidase (DAO) activity, and D-lactate (D-LA) level in IPEC-J2 cells exposed to H_2_O_2_. (**D**) Detection of apoptosis based on Annexin V-FITC staining and PI staining. Flow-cytometric analysis of apoptotic cells. (**E**) Quantification of apoptosis based on flow cytometry data. (**A**–**C**,**E**) Data were analyzed by one-way ANOVA and multiple comparisons with Duncan’s multiple range test. Data are presented as mean ± SEM with three independent experiments. n = 3 means three independent experiments. The differences between two groups were considered significant at * *p* < 0.05 and ** *p* < 0.01. ns indicates no difference between two groups.

**Figure 3 antioxidants-13-00468-f003:**
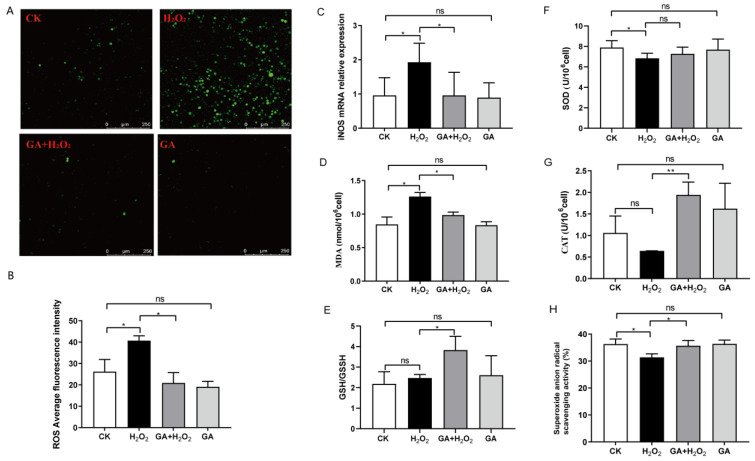
Effects of glycyrrhetinic acid (GA) on redox status of oxidative stress-induced IPEC-J2 cells. (**A**) The production of ROS was assessed by DCFH-DA staining and fluorescence microscopy. (**B**) ROS production was calculated. (**C**) The mRNA expression of *iNOS* in IPEC-J2 cells. (**D**–**H**) Effect of GA on MDA level, GSH/GSSG, SOD, CAT, and superoxide anion radical scavenging activities in IPEC-J2 cells. (**B**–**H**) Data were analyzed by one-way ANOVA and multiple comparisons were analyzed with Duncan’s multiple range test. Data are presented as mean ± SEM with three independent experiments. n = 3 means three independent experiments. The differences between two groups were considered significant at * *p* < 0.05 and ** *p* < 0.01. ns indicates no difference between two groups.

**Figure 4 antioxidants-13-00468-f004:**
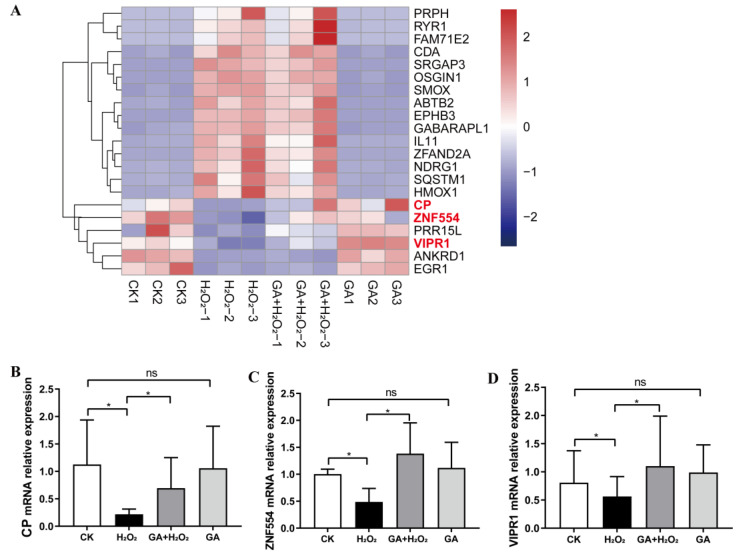
Differentially expressed up- and downregulated genes based on heat map analysis. (**A**) The heatmap of gene expression; the red, white, and blue colors represent the upregulated, unchanged, and downregulated differentially expressed genes in the GA + H_2_O_2_ vs. H_2_O_2_ group. (**B**–**D**) RT-qPCR validation of differentially expressed genes. (**B**–**D**) Data were analyzed by one-way ANOVA and multiple comparisons with Duncan’s multiple range test. Data are presented as mean ± SEM with three independent experiments. n = 3 means three independent experiments. The differences between two groups were considered significant at * *p* < 0.05. ns indicates no difference between two groups.

**Figure 5 antioxidants-13-00468-f005:**
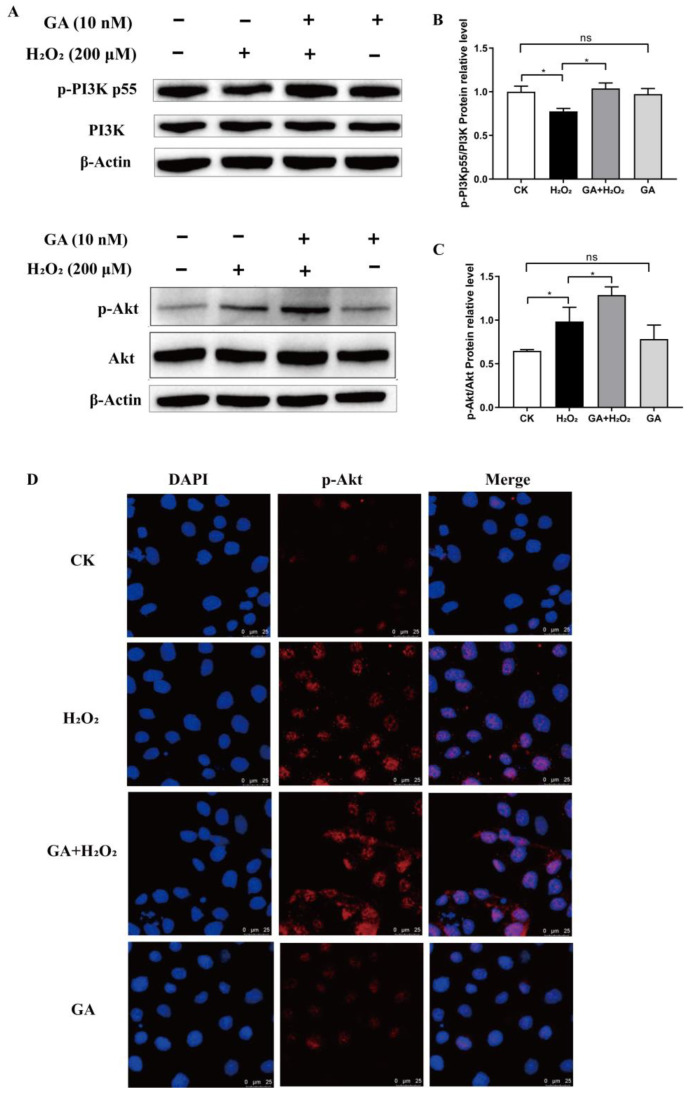
Glycyrrhetinic acid (GA) regulates the PI3K/Akt signaling pathway in IPEC-J2 cells. (**A**) The protein expression of PI3K, p-PI3K, Akt, and p-Akt was measured by Western blot analysis with specific primary antibodies, and β-actin was used as a loading control. (**B**,**C**) Quantification of the relative protein expression was performed by densitometric analysis. (**D**) The expression level of p-Akt was assessed by immunofluorescence analysis. Representative images were taken at 200× magnification. DAPI was used as a nuclear counterstain. (**B**,**C**) Data were analyzed by one-way ANOVA and multiple comparisons with Duncan’s multiple range test. Data are presented as mean ± SEM with three independent experiments. n = 3 means three independent experiments. The differences between two groups were considered significant at * *p* < 0.05. ns indicates no difference between two groups.

**Figure 6 antioxidants-13-00468-f006:**
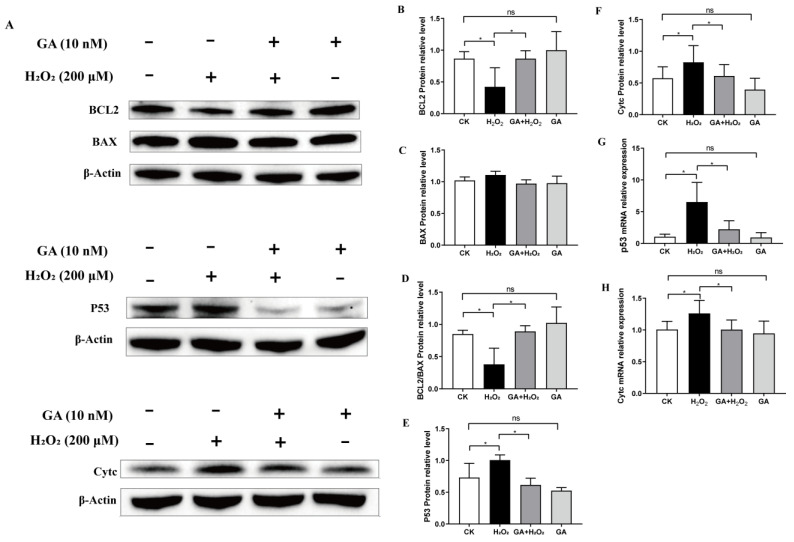
Glycyrrhetinic acid (GA) prevents oxidative stress-induced apoptosis by regulating the downstream targets of PI3K in IPEC-J2 cells. (**A**–**F**) The protein expressions of p53, Cytc, BCL2, and BAX were measured by Western blotting with specific primary antibodies, and β-Actin was used as a loading control. (**G**,**H**) The mRNA expressions of *p53* and *Cytc*. Quantification of the relative protein expression was performed by densitometric analysis. (**B**–**E**) Data were analyzed by one-way ANOVA and multiple comparisons with Duncan’s multiple range test. Data are presented as mean ± SEM with three independent experiments. n = 3 means three independent experiments. The differences between two groups were considered significant at * *p* < 0.05. ns indicates no difference between two groups.

**Figure 7 antioxidants-13-00468-f007:**
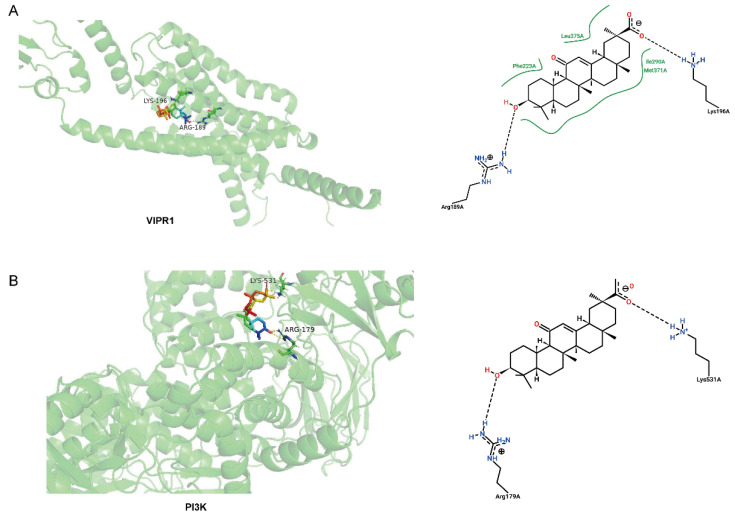
Molecular binding of glycyrrhetinic acid (GA) with VIPR1 and PI3K. (**A**) The 3D structure interactions of GA with VIPR1. (**B**) The 3D structure interactions of GA with PI3K.

**Table 1 antioxidants-13-00468-t001:** Binding energy and docking amino acids sites of glycyrrhetinic acid (GA) and potential target proteins.

Protein	Residue	AA	C	Distance H-A	Binding Energy/(kcal mol^−1^)
VIPR1	189A	ARG	C3	1.9	−6.58
	196A	LYS	C30	1.7	−6.35
PI3K	179A	ARG	C3	1.9	−5.39
	531A	LYS	C30	1.8	−5.39

## Data Availability

The data presented in this study are available on request from the corresponding author.

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
