# Peer review of "18Beta-Glycyrrhetinic Acid Attenuates H2O2-Induced Oxidative Damage and Apoptosis in Intestinal Epithelial Cells via Activating the PI3K/Akt Signaling Pathway"

_antioxidants, 2024, doi:10.3390/antiox13040468_

Round 1

Reviewer 1 Report (Previous Reviewer 3)

I do not find any major issues with the revised manuscript.

The authors have duly revised the manuscript and addressed the reviewers concerns.

Reviewer 2 Report (Previous Reviewer 2)

The authors well responded to my previous comments. Thank you for this revised version.

Nothing to add respect this version of the article.

This manuscript is a resubmission of an earlier submission. The following is a list of the peer review reports and author responses from that submission.

Round 1

Reviewer 1 Report

Comments and Suggestions for Authors

The objective of the manuscript (Ms) by Ma, C. et al, was to evaluate the protective activity of 18β- glycyrrhetinic acid (GA) in oxidative stress-induced apoptosis and identify the underlying mechanisms. The authors showed that GA partially protects intestinal porcine epithelial cells (IPEC-J2) cells from H2O2-induced oxidative stress by increasing the cell viability and decreasing ROS production and cell apoptosis. These data are rather confirmatory as at least other two papers showed very similar data in HepG2 and Schwann cells. However, in search of possible mechanisms underlying GA effects, the authors performed RNA seq analysis and detected several genes differentially modulated by H2O2 in IPEC-J2 cells, but only few genes partially modulated by GA, including a membrane receptor VIPR1. The in-silico docking demonstrated that GA may bind to VIPR1 and PI3K, thus activate the anti-apoptotic pathway PI3K/AKT.  These are novel and interesting, but incomplete findings and would need a functional readout that the authors do not provide. Importantly, the close examination of original Western blots revealed that the author did not follow the standard rules that are required for the data analysis and presentation. In addition, the paper has many other problems as pointed out below:

Major points:

1)      The examination of presented Western Blots revealed several problems: Loading controls in most of the figures are from different blots than other presented bands, which is an unacceptable practice. In addition, several other problems can be pointed out. For example, B-actin in Fig, 5A is on a clearly separate blot with double protein ladder, and other blots (ph-p85, p85 PI3K, ph-Akt and Akt) are also on different blots and contain one or no marker lane. In addition, ph-p85 and p85 PI3K, as well as ph-Akt and Akt are on completely different blots, thus phosphorylated and unphosphorylated bands are on different blots. In addition, ph-p85 and p85 PI3K contain a similar pattern of bands (strong upper band and weaker lower band), but in the original image 1 ph-p85 is a lower band, while p85 PI3K is the upper strong band. This means that the authors changed even the acrylamide concentrations for analysis of p85 phosphorylated and unphosphorylated. Several other irregularities could be pointed out on other images, beside non-corresponding loading controls.

2)      The theoretical docking of GA to VIPR is not a proof that GA really binds to this receptor and mediates PI3K/akt signaling. The authors should show other evidence to confirm the GA binding to VIPR by one of existing experimental methods (CETSA, radioligand binding or others…). Silencing of VIPR by siRNA or other methods to eliminate VIPR expression should verify if the GA binding to this receptor mediates PI3K/Akt activation.

3)      The authors identified VIPR as upregulated mRNA by RNA seq, but they did not show that the protein is induced by GA.

4)      Most of graphs show only one representative experiment out of 3 experiments and the data are presented as the mean ± S.D. of triplicate cultures, instead of the mean+/- SEM of more experiments. While it is occasionally acceptable for one or two experiments, it should not be a practice, therefore the authors should calculate and present the data as the mean +/- SEM of more independent experiments. In addition, the Methods section indicates that: “each group contained five replicates per trial”, but all the figure legends indicate n = 3.

5)      In “2.6 Redox Status Analysis” there is no reference to the method/kit used. In fact, it is not easy to perform the measurements of MDA levels and GSH/GSSG ratio as well as SOD and CAT activity and not all commercially available kits are reliable. The authors should provide a description of the methods and kits used and provide the raw data obtained (as supplementary data), to convince the readers that assays were correctly performed. 

6)      RNA seq analysis identified 21 genes strongly modulated by H2O2 in IPEC-J2 cells, but only 3 genes weakly modulated by GA. This is rather inconsistent with the significant antioxidant and anti-apoptotic effects of GA presented in functional assays and protein expression analysis in Figs. 1 to 3 and 5 to 6. The authors should provide a convincing explanation of this incongruency in the Discussion.

English comments:

The text remains comprehensible in most of the cases, but English usage in the Ms should be revised by a professional editing help or a native speaker, as it is full of style and grammar errors and few examples are pointed out in minor points.

Minor points:

1)      CP mentioned 3 times should be defined as Ceruloplasmin at the first mention and the same is for VIPR1 and other abbreviated genes/proteins described in the text.

2)      The cellular function of VIPR1 should be briefly described in the discussion with relevant citations.

3)      The abc system for the statistical significance is rarely used and might be confusing. The reviewer suggests more transparent indicators, like connecting lines with asterisks.

4)      Several references are wrong: References regarding VIPR1-induced stimulation of PI3K/akt pathway are missing. The indicated 28 and 29 ref.s do not even mention VIPR1!

5)      The statistical post-tests should be indicated in the Methods section, or in each legend if different post-tests were used.

6)      The description of methods used is often incomprehensible, mainly due to bad English. Just a few examples:  

1.       Lines 91-93: “the group of GA pre-treatment with 24 h exposure to 200 uM H2O2 for 8 h (GA+H2O2), and the group of GA pre-treatment with 24 h (GA), and the same as below in our study. ??? The authors probably wanted to write: “GA pre-treatment for 16 h followed by co-treatment with 200 uM H2O2 for 8 h (GA+H2O2), and the group of GA treatment for 24 h (GA), …”

2.       Line 93: “…and the same as below in our study.”: totally unclear what the authors meant by these words

3.       Line 123-124: “The cell culture supernatant was collected to detected the malondialdehyde (MDA), …”. It should be “…. to detect ….

4.       Line 126-127: “Total RNA was using the RNA Nano 6000 Assay Kit of the Bioanalyzer 2100 system 126 (Agilent Technologies, CA, USA) to assess.”It probably it should be: “Total RNA was assessed by using the RNA Nano 6000 Assay Kit of the Bioanalyzer 2100 system (Agilent Technologies, CA, USA).”

5.       Line 161-164: “… cells were cultured in 24-well plates at a concentration of 1 × 105 cells/well to attach for 24 h, then treated with …… In the culture plate, the slides with climbed cells were washed in PBS for 5 minutes,.…” It probably it should be: “… cells were plated on glass slides in 24-well plates at a density of 1 × 105 cells/well for 24 h, then treated with …Subsequently, the plates with slides were washed in PBS for 5 min., ….”

7)      Many other sentences should be corrected throughout the entire Ms as roughly 1 in 2 need to be corrected.

Comments on the Quality of English Language

The text remains comprehensible in most of the cases, but English usage in this Ms should be revised by a professional editing help or a native speaker, as it is full of style and grammar errors and few examples are pointed out in minor points (See above minor points 6 and 7).

Reviewer 2 Report

Comments and Suggestions for Authors

Dear authors, the manuscript is good but needs to be improved before publication. Here are my comments. 

1. P-2 material an experiment. The concentration of used DMSO has to be mentioned.  2. P-3 Apoptosis analysis, Please check the grammar of the following text. "And then, the 200 μL 1 × binding 106 buffer and 5 μL Annexin V-FITC were added to 100 μL IPEC-J2 cell suspension at room 107 temperature in black for 15 min". 3. P-5 Results, check the grammar of the following text. "We first investigate the effect of 200 μM H2O2 on the viability of IPEC-J2 cells using CCK-8 assay and found that the viability was significantly reduced to 70% upon exposure to 200 μM H2O2 after 8 h incubation compared to those without H2O2 treatment". 4. In each figure caption the statistical method has to be clarified (one-way ANOVA or two-way ANOVA ) since there are 2 independent variables. 5. P-9 lines 332, 333, and 334 contain inaccuracies. Comments on the Quality of English Language

Minor editing of English language required

Reviewer 3 Report

Comments and Suggestions for Authors

In this study, the authors evaluated the protective activity of 18β-GA (18β-glycyrrhetic acid) in oxidative stress-induced apoptosis in intestinal epithelial cells and identify the molecular mechanisms involved. The authors showed that GA protects intestinal porcine epithelial cells (IPEC-J2) cells from H2O2-induced oxidative stress by increasing the cell viability, inhibiting the production of intracellular ROS and decreasing apoptotic proteins/gene expression (specifically Cytochrome C and p53). Interestingly, molecular docking analysis predicted the binding of GA to VIPR1, a master membrane receptor, which further leads to the activation of PI3K/AKT signaling. Overall, these studies have provided mechanistic insights into the potential therapeutic role of GA in preventing diseases associated with oxidative stress, such as acute/chronic intestinal and/or liver disease. However, the authors should consider the following minor suggestions:

1. How did the authors choose the doses of GA in the current study? This is relevant in lieu of a recent study of Malekinejad M et al (Eur J Nutr. 2023 Apr;62(3):1561) showing that at higher doses, GA exhibits detrimental effects on intestinal epithelial cells.

2. Figure legends: Please clarify which groups are compared to achieve the p value in all the figures shown.

3. Proofread the manuscript carefully for typo and grammatical errors.

Comments on the Quality of English Language

Some grammatical and typo errors noted
